# Clinical Pharmacology of Factor XI Inhibitors: New Therapeutic Approaches for Prevention of Venous and Arterial Thrombotic Disorders

**DOI:** 10.3390/jcm11216314

**Published:** 2022-10-26

**Authors:** Elena Campello, Paolo Simioni, Paolo Prandoni, Nicola Ferri

**Affiliations:** 1Department of Medicine, University of Padova, 35128 Padova, Italy; 2Arianna Foundation on Anticoagulation, 40138 Bologna, Italy

**Keywords:** anticoagulation, venous thromboembolism, atrial fibrillation, thrombosis, bleeding

## Abstract

Bleeding is the dominant adverse event of anticoagulation and often discourages many patients and physicians from starting treatment with anticoagulant drugs. The fact that factor (F)XI deficiency is associated with a mild bleeding phenotype and that FXI knockdown or inhibition in different animal models reduced the occurrence of thrombotic events in response to injury suggests that FXI is more important for the coagulation propagation and thrombotic process than for the overall hemostasis. The aim of this review is to summarize clinical pharmacology and evidence from phase 2 clinical trials on efficacy and safety of drugs directed against FXI for the treatment and prevention of thrombosis. Inhibition of FXI or FXIa has been proven to be effective in phase 2 studies at preventing venous thromboembolism (VTE) in patients undergoing total knee arthroplasty, or for prevention of major adverse vascular events in patients with end-stage kidney disease undergoing hemodialysis or as adjuncts to antiplatelet therapy for prevention of recurrent ischemic events in patients with acute myocardial infarction or non-cardioembolic stroke. Should the efficacy of FXI inhibitors as anticoagulant without impairing the hemostasis be proven in phase 3 randomized clinical trials, it would provide an innovative therapeutic option.

## 1. Introduction

Anticoagulation is the most important component of the management strategy for several common medical conditions. Particularly, it is indicated for the prevention of recurrent thrombosis in patients with venous thromboembolism (VTE), which includes deep venous thrombosis and pulmonary embolism, and ischemic stroke due to atrial fibrillation (AF) [1,2,3]. Considering that VTE occurs in 1 to 2 individuals per 1000 each year and that AF is the most common sustained cardiac arrhythmia in adults [2,3], the use of anticoagulation in everyday clinical practice is very frequent. If on one hand anticoagulant therapy is very effective at preventing recurrent thrombosis, it is associated with an increased frequency of bleeding complications [4]. Major bleeding such as intracranial hemorrhage (ICH) or massive gastrointestinal hemorrhage discourage many patients and physicians from starting treatment with anticoagulants [4]. According to a review, in 23 randomized clinical trials in patients with AF using vitamin K antagonists (VKAs), the median rate of major bleeding was 2.1 per 100 patient years [5]. With the advent of direct oral anticoagulants (DOACs), the risk of major bleeding was reported about 30% lower than with VKAs [4,6]. Particularly, in a systematic review of observational studies in AF, the relative risk reduction for major bleeding with apixaban vs. warfarin was 38%, with dabigatran vs. warfarin 35%, and with rivaroxaban vs. warfarin 46% [7]. However, no studies comparing different DOACs head-to-head are available and the corresponding absolute risk differences are likely to be small (i.e., 1.7%, 1.4%, and 1.2%, respectively) [4]. Importantly, there is a 50% reduction of ICH and fatal bleeds with DOACs vs. VKAs [6,8], but again the absolute reduction is limited to 2 intracranial bleeds and 1 fatal bleed per 1000 patients per year [4]. As for neurological function after ICH, it does not seem to differ between DOACs and AVKs [9,10]. The most common type of major bleeding with oral anticoagulants is from the gastrointestinal (GI) tract [4]. In trials enrolling patients with AF, GI bleeding was more frequent in patients taking DOACs vs. VKAs and the difference was mainly due to a higher risk for major GI bleeding with dabigatran and rivaroxaban vs. VKAs, but not with apixaban and edoxaban vs. VKAs [4]. Many unmet needs remain in the field of thrombosis management, both for AF and VTE patients, even in the DOACs era (Table 1). All the currently available anticoagulant drugs antagonize activated factor X (FXa) and/or thrombin, either directly or indirectly [11]. These two proteases play an essential role for the overall hemostasis and therefore administration of current anticoagulants coincide with an increased risk of bleeding. Efforts to find a factor that upon antagonization prevents thrombosis without impairing the overall hemostasis have resulted in the identification of coagulation factor XI (FXI) as a possible ideal therapeutic target [11,12].

The aim of this review is to summarize pharmacology and current clinical evidence on efficacy and safety of drugs directed against FXI/FXIa in treatment and prevention of thrombosis.

## 2. Targeting Factor XI

FXI directly activates the extrinsic pathway through multiple mechanisms interacting with several enzymatic substrates. The exclusive involvement of FXI in the “contact” pathway has been excluded after the discovery that the same factor can be activated by thrombin downstream of the extrinsic pathway [1]. Thus, the main substrate for FXIa is the contact pathway factor FIX, however, FXI has also been shown to activate FX in vitro and to promote thrombin generation by activating the cofactors FVIII and FV [2,3]. FXIa was also known to shorten the clotting time in a FIX independent manner [4]. Finally, FXIa promotes activation of the extrinsic pathway through proteolysis of tissue factor pathway inhibitor (TFPI) [5], a Kunitz-type protease that inhibits the TF/FVIIa/FXa complex [6]. Additionally, the presence of platelets is required for FXIa to support hemostasis through the extrinsic pathway [7,8]. Thus, the role of FXI on coagulation cascade is very complex and likely not completely elucidated.

FXI deficiency is associated with a mild bleeding phenotype often called hemophilia C. Particularly, muscles or joints and life-threatening ICH or gastrointestinal bleeding seen in hemophilia A or B do not characterized FXI deficiency [13,14]. On the other hand, epidemiological data and studies in animals support the importance of FXI in thrombosis. In fact, FXI-deficient individuals appear to have reduced incidences of VTE and ischemic stroke compared with the general population, whereas those with high FXI levels carry more than twice the risk of VTE [11,15,16]. Additionally, FXI knockdown or inhibition in different animal models reduced the occurrence of thrombotic events in response to injury [17,18]. These observations suggest that FXI is more important for the thrombosis process than for the overall hemostasis. During the thrombotic occurrence, thrombus formation appears to be triggered by tissue factor (TF) exposed on altered vascular endothelium or expressed on leukocytes, extracellular microvesicles, or injured endothelial cells [11,12,19,20]. However, the capacity of the FVIIa/TF complex to propagate intraluminal thrombus growth may be limited once the thrombus enlarges and extends beyond the TF source at the site of the injured vessel wall [11]. Indeed, FXI appears to drive thrombus growth. Thrombin is the major activator of FXI in the thrombosis process. Additionally, polyanions that are likely to be present in a growing thrombus, including chromatin extruded from activated neutrophils (neutrophil extracellular traps) or inorganic polyphosphates released from activated platelets or microvesicles released from red blood cells or activated platelets, can activate FXII and support additional FXI activation by FXIIa [11,12,21]. Furthermore, in situations where thrombosis is triggered by exposure of blood to artificial surfaces (e.g., cardiopulmonary bypass, hemodialysis, central venous catheters), thrombin generation may be primarily initiated by FXIIa activation of FXI, rather than by the FVIIa/TF complex [11,21]. Regardless of the mechanism of activation, FXI appears to be the optimum target to control thrombin generation and thrombus enlargement.

## 3. Clinical Pharmacology of Anti FXI and FXIa

The new era of pharmacology brought pharmaceutical companies to utilized additional approaches, beyond the classic small chemical molecules, to control the coagulation cascade, such as monoclonal antibodies (mAbs) and antisense oligonucleotides (ASO). Here we will focus on FXI and FXIa inhibitors that will be potentially approved as new antithrombotic drugs (Table 2).

### 3.1. mAbs anti FXI and FXIa

#### 3.1.1. Abelacimab

##### Mechanism of Action

Abelacimab is a monoclonal antibody capable to bind and inhibit both FXI and FXIa [40]. The Fab portion of abelacimab exhibited high binding affinity to FXI and FXIa and inhibited the 50% of FXIa activity at concentration of 2.8 nM. This effect was shown to be very specific with no inhibitory effect on other human serine protease-type coagulation factors, including factor VIIa, factor IXa, factor Xa, FXIIa, thrombin, and its closest homolog plasma kallikrein [40]. Abelacimab binds with high affinity to the catalytic domain of human FXI and FXIa (Kd of 1.3 ± 0.3 pM and 4.7 ± 2.1 pM, respectively) [40]. The analysis of the X-ray structure of the Fab portion of abelacimab in complex with the FXIa catalytic domain revealed that the antibody traps and stabilizes, an inactive conformation, the protease. By meaning of this activity, abelacimab prolonged the clotting timer, determined by using the aPTT assay, and reduced the amount of thrombin in human plasma in a concentration dependent manner [40].

These in vitro studies demonstrated that abelacimab inhibited FXIa, activated by FXIIa in the intrinsic pathway, as well as by thrombin, but did not interfere with the initial thrombin burst triggered by the extrinsic pathway [40]. Thus, abelacimab prevents the activation of circulating (zymogen) FXI and inhibits any FXIa that has already been formed.

##### Pharmacokinetics

The pharmacokinetic (PK) properties of abelacimab were assessed in humans after a single s.c. administration of ascending doses (5 mg up to 240 mg) [40]. Abelacimab exposure is dose proportional within the studied dose range with moderate intersubject variability [40]. The median time to-peak abelacimab plasma concentrations ranged from 7 to 21 days and were dose independent. However, levels adequate to prolong aPTT more than two-fold were achieved by 24 h. The terminal elimination half-life (T_1/2_) ranged from 20.1 to 28.6 days (580 h; Table 3). Severe obesity was associated with modest decreased abelacimab C_max_ and AUC [40].

To explore the PK and PD properties of abelacimab following different routes of administration, two additional studies were conducted with single i.v. administration of the drug in healthy volunteers and with a multiple once-monthly s.c. administration in patients with AF [22]. After single i.v. administration the T_max_ of abelacimab ranged from 1.75 to 2.0 h (after start of the 1-h infusion) and was found independent from the dose [22]. The systemic exposure (C_0_) increased proportionally in the dose range of 30–150 mg and more than dose proportional for AUC_inf_. The half-life time of elimination (T_1/2_) ranged from 595 and 709 h (25–30 days) (Table 3) [22]. The volume of distribution (Vd) is between 5 L and 13.3 L after i.v. and s.c. administration, respectively. Severe obesity (median BMI 44.3 kg/m^2^) was associated with a moderate decrease (30–45%) in abelacimab exposure (C_0_ and AUC) and Vd (8.29%).

##### Pharmacodynamics

After 1 h post i.v. infusion abelacimab binds >99% of FXI across all doses (30 mg up to 150 mg). At the dose of 150 mg of abelacimab, the reduction of free FXI concentration remained sustained until 60 days post administration. By day 106, the inhibitory effect of abelacimab on FXI returned to levels consistent with those observed at baseline [22]. Reductions in free FXI levels were associated with a rapid and sustained prolongation of aPTT. The 2-fold aPTT time was observed between 1 h and 5 days post infusion [22]. In particular, at the dose of 150 mg, the peak of 2-fold aPTT prolongation was observed at 48 h and returned to baseline at day 106 [22].

The repeat monthly s.c. administration of abelacimab led to a robust and sustained reduction in free (unbound) FXI levels compared to baseline and placebo (Figure 1) [22]. This inhibition was complete (≈99%) after the first and at 30 and 60 days until day 90 at the dose of 180 mg, while some rescue was observed at the dose of 120 mg (Figure 1) [22]. During the washout/follow-up period, concentrations of free FXI in both the abelacimab 120 and 180 mg dose groups returned to near baseline levels up to day 110 (end of study visit).

The effect on FXI produced a 1.8 to 2-fold prolongation of aPTT after 11 days post injection for both the abelacimab 120 and 180 mg dose groups (Figure 1). The aPTT decline to 1.6-fold before the second dose for both 120 mg and 180 mg. Thus, s.c. administration of abelacimab at 180 mg every 30 days seems to have a sustained and complete inhibition of FXI with a significant 2-fold increase of aPTT time [22].

#### 3.1.2. Osocimab

##### Mechanism of Action

Osocimab is a fully human IgG1 mAb generated using phage display technology. Crystal structure analysis demonstrated that osocimab inhibit FXIa through an allosteric mechanism of action based on a direct and specific binding to a region adjacent to the FXIa active site [41]. Osocimab binds specifically to human FXIa with an EC_50_ of 0.2 ± 0.02 nM [41]. This interaction leads to a structural rearrangements and inhibition of FXIa activity. More in detail, FXI has a unique homodimeric structure and it is well accepted that, via a transactivating mechanism, binding of FXIIa or thrombin to one FXI subunit leads to the activation of the other monomer [42]. Osocimab, by binding to a loop region, only accessible in the activated form of FXI, inhibits one half of the activated dimer and thus blocking its catalytic activity. Indeed, osocimab does not bind inactive FXI. The inhibition of FXIa-mediated activation of FIX (the natural substrate of FXIa) in human plasma occurs with an IC_50_ of 16 ± 0.02 nM [41]. Osocimab was shown to be very selective for FXI without any significant effect against other serine proteases, such as thrombin, FXa, FVIIa, trypsin, plasmin, tissue plasminogen activator (t-PA), and kallikrein [41]. Osocimab prolong the clotting time of human blood in a concentration-dependent manner and the concentration required to double the aPTT was 0.14 µM. Thrombin generation, triggered with phospholipids and low concentration of tissue factor (TF), was inhibited with an IC_50_ value of 0.035 µM.

##### Pharmacokinetics

In a phase 1 study, the PK profile of osocimab was evaluated in healthy volunteers after a single dose as a 60-min i.v. infusion [24]. There was a dose-dependent increase of osocimab exposure (AUC_0-last_ and C_max_) after administration of nine doses (0.015, 0.06, 0.15, 0.3, 0.6, 1.25, 2.5, 5, or 10 mg/kg). The drug was undetectable below the dose of 0.06 mg/Kg. Variability in AUC_0-last_ and C_max_ was moderate with geometric CVs. of 12.7–34.5% and 11–20.3%, respectively (Table 4). The T_max_ of osocimab was between 1 and 4 h after the start of the infusion, and the elimination half-life (t_1/2_) was approximately 30–44 days (729–1050 h). The clearance and Vd were in the range 0.0054–0.0787 L/h and 5.4–8.7 L, respectively (Table 4). An exploratory ANOVA analysis, normalized by dose or by dose and body weight, did not suggest clear dose proportionality of exposure parameters [24].

##### Pharmacodynamics

The i.v. administration of osocimab produced a dose-dependent increase in aPTT and reduction of apparent FXI activity [24]. A sustained elevation of aPTT above baseline was observed at the dose of 1.25 mg/Kg up to day 55 post infusion (Figure 2). At the dose of 10 mg/kg, osocimab increases aPTT relative to baseline with the conventional and kaolin-triggered methods by 1.85 and 2.17, respectively (Figure 2). Osocimab inhibits FXI activity almost immediately after drug administration and gradually returned baseline levels after approximately 55 days [24]. Whole blood rotational thromboelastometry showed a dose-dependent elevation of clotting time with a duration of at least 144 h [24]. Importantly, bleeding times, evaluated up to 7 days after drug administration, did not increase and were similar between patients who received osocimab or placebo [24].

#### 3.1.3. BAY 1831865

##### Mechanism of Action

Since abelacimab binds directly to the FXI catalytic domain [23] and osocimab to a region adjacent to the active site providing an allosteric inhibitory activity [24], an antibody combining FXI zymogen inhibition with active site-focused inhibition may have distinct properties. BAY 1831865 is a humanized, sequence-optimized, mAb that binds specifically to the apple domain 3 of FXI [26]. This domain binds several ligands, including FIX and FXIIa. Thus, BAY 1831865 hinders the binding of FIX with FXIa and thus the generation of FIXa and inhibits FXI activation to FXIa by blocking the FXIIa-mediated activation of FXI.

##### Pharmacokinetics

The PK profile of BAY1831865 has been evaluated in healthy volunteers after dose escalating i.v. injection and a single s.c. administration. Following i.v. administration, there was a dose-dependent increase in plasma concentration of BAY 1831865 (C_max_) without a significant deviation from dose-proportionality [26]. However, the t_1/2_ increased with the dose, from 28.4 h at 7 mg until 208 h at 150 mg, suggesting a potential accumulation of the drug. This was reflected by a consistent increase in AUC/D, with values ranging from 12.3 h/L to 72.9 h/L after 7 mg and 150 mg of infused dose. Median T_max_ was similar for all i.v. doses, ranging from 1.0 h to 2.0 h after the start of infusion. As expected, a dose-dependent decrease of clearance (CL) was observed, from 0.0815 L/h at 7 mg until 0.0137 L/h at 150 mg [26].

The AUC of 150 mg BAY 1831865 after s.c. administration was about half of the same dose after i.v. infusion (5160 h µg/mL vs. 10 900 h ּ µg/mL), and C_max_ was reduced by almost 80% (10.9 mg/L vs. 48.3 mg/L) (Table 5), while Vd was almost double after s.c. injection (4.17 L vs. 9.10 L). The T_max_ after s.c administration was significantly longer than after i.v. (96.0 h vs. 1–2 h), while the t_1/2_ was similar (217 h vs. 208 h). The absolute bioavailability of BAY 1831865 150 mg s.c. vs. 150 mg i.v. was 47.2% (Table 5).

##### Pharmacodynamics

After i.v. injection, BAY 1831865 produces a dose-dependent increases and prolongation in aPTT. The maximal effect was observed at the dose of 150 mg i.v. with 3.11-fold increase of aPTT compared to placebo (1.05) (Figure 3). The interindividual variability of this effect was low (CV range, 3.2–10.0%). Onset of action after i.v. injection of BAY 1831865 was rapid, with increases in aPTT within 1 h for doses of 7 mg or greater. Duration of aPTT prolongation was dose-dependent, with ratio-to-baseline values returning to baseline levels between 2 days (7 mg) and 55 days (75 mg and 150 mg) after the infusion (Figure 3). Differently to i.v. administration, aPTT increased slowly with BAY 1831865 150 mg s.c. until about 5 days after dosing (Figure 3). BAY 1831865 150 mg s.c. produced a 2.78 increase of aPTT (Figure 3). aPTT remained elevated at 27 days and returned to baseline at 55 days after s.c. administration (Figure 3).

Dose-dependent inhibition of FXI activity (Figure 3) and FXIa activity was observed with increasing i.v. doses of BAY 1831865. After administration of any i.v. dose, the onset of action was rapid with a significant inhibition both FXI and FXIa activity. FXI activity returned to baseline between 13 days (3.5 mg) and 55 days (35 mg, 75 mg, and 150 mg). For FXIa activity, values returned to baseline after 13 days with the 3.5 mg i.v. dose and were almost, but not fully, restored to baseline during the observation period (55 days) at the dose of 150 mg (0.82 vs. basal) [26].

Differently from the i.v. injection, the s.c. administration of BAY 1831865 150 mg reduced the FXI and FXIa activity slowly, and the minimum activity of both variables observed after 3 days (Figure 3). Baseline FXI activity values were attained at 55 days, but FXIa activity remained below baseline at this time. Clotting time, performed using rotational thromboelastometry, was increased in a dose-dependent manner, with a ratio-to-baseline values ranging from 1.2 to 3.4 after administration of 3.5 mg i.v. and 150 mg i.v., respectively. The onset of effect after s.c. administration on clotting time was slower compared to i.v., with ratios to baseline value equal to 2.9 [26].

#### 3.1.4. AB023/Xisomab 3G3

##### Mechanism of Action

AB023 is a recombinant, humanized mAb that binds the apple 2 domain of FXI and prevents its activation by FXIIa. However, AB023 does not affect the activation of FXI by thrombin, nor does it prevent FXIa from activating FIX. Therefore, even though AB023 binds to FXI, it functions as a FXIIa inhibitor. Consistent with its mechanism of action, AB023 prolongs the aPTT in a concentration-dependent manner but not the prothrombin time in human plasma.

##### Pharmacokinetics

Phase 1 trial reveals important PK information of AB023 after bolus administration of four ascending doses (0.1, 0.5, 2.0, and 5.0 mg/kg) [27]. After the injection of lowest doses (0.1 and 0.5 mg/Kg), AB023 was detectable in plasma for 120-h, while at 2.0 mg/Kg and 5.0 mg/Kg unbound AB023 was present throughout the entire sampling interval (672 h). T_max_ of AB023 increased with the dose, from 0.08 h for the 0.1 mg/kg up to 1 h for the 5 mg/kg dose level. The exposure to the mAb increased slightly more than a dose-proportional manner from the 0.5 mg/kg to 5.0 mg/kg. T_1/2_ values appeared to increase with increasing AB023 dose, ranging from 1.3 h at 0.1 mg/kg up to 121.5 h following the 5.0 mg/kg dose (Table 6). CL and Vd were also dose dependent. These data indicate that the short half-life of free AB023 at lower doses may due to the rapid binding of the mAb to FXI. In contrast, when AB023 was given in doses sufficient to saturate FXI, the long half-life of free AB023 can be explained by its slow clearance via the neonatal Fc receptor [43].

##### Pharmacodynamics

As expected from the mechanism of action, AB023 has been shown to prolong the aPTT in a dose-dependent manner [27]. The dose of AB023 required to prolong the aPTT by approximately 2-fold was 0.5 mg/kg, effect that last 7 days. The aPTT prolongation reached a maximum value at ≥0.5 mg/kg, dose capable to saturate the binding of all free FXI. However, at the dose of 5 mg/kg, the effect on aPTT last for more than 42 days (Figure 4). Thus, at appropriate dose, AB023 could potentially be administered once every 6 weeks, indeed the aPTT values of most subjects treated with 2.0 mg/kg of AB023, and all subjects at the dose of 5.0 mg/kg returned to baseline after 672 h (Figure 4).

Taken into consideration the mechanism of action of AB023 that blocks the FXIIa-mediated activation of FXI, this mAb could be utilized for preventing clotting formation on blood-contacting medical devices (hemodialysis catheters, central venous catheters, left ventricular assist devices, or mechanical heart valves) or extracorporeal circuits. In addition, due to its long-lasting activity (more than one month), AB023 would be an attractive alternative to warfarin for prevention of clotting on mechanical heart valves or left ventricular assist devices.

### 3.2. Small Molecules Anti FXI and FXIa

Pharmaceutical studies aim at identifying potent FXIa inhibitors suitable for oral administration led to the identification of small molecules named asundexian and milvexian that are currently in clinical development.

#### 3.2.1. Asundexian

##### Mechanism of Action

Asundexian (BAY 2433334) is a chemically synthesized molecule, which potently inhibits FXIa in a reversible manner. The IC_50_ for in vitro inhibition of FXIa is equal to 1.0 ± 0.17 nM [29] and FXIa activity in human plasma with an IC_50_ of 0.14 ± 0.04 μM. Asundexian showed an excellent selectivity for FXIa without any significant inhibitory action on other proteases, including thrombin, FXa, FIXa, FXIIa, FVIIa, urokinase, t-PA, plasmin, activated protein C, trypsin, and chymotrypsin [29]. Asundexian produced a concentration-dependent prolongation of aPTT in human plasma and doubled the clotting time (aPTT) at concentration of approximately 2µM [44]. By contrast, asundexian had no significant effect on PT.

##### Pharmacokinetics

The PK profile of asundexian has been evaluated in a phase 1 clinical trial in healthy volunteers randomized to receive a single oral dose of 5–150 mg as oral solution or immediate-release tablets or placebo. In addition, the effect of food has also been after administration of 5 mg tablet of asundexian [29].

The T_max_ of asundexian of oral solution and tablets was 1 h and 2.5–4 h, the t_1/2_ was 14.2 to 17.4 h, and CL was 3.19 to 4.21 L/h, respectively (Table 7). There was no deviation from a dose-proportional increase in asundexian exposure after administered as oral solution or tablets. High-fat, high-calorie breakfast showed a minimal effect on the bioavailability of the five 5-mg of asundexian tablets. AUC decreased by 12.4% and C_max_ by 31.4% in the fed vs. the fasted state.

##### Pharmacodynamics

After oral administration, asundexian produces a rapid and dose-dependent increase in aPTT at all doses. The maximal ratio to baseline in aPTT was 2.12, at the dose of 25 mg of oral solution, corresponding to an absolute value of 36.6 s (Figure 5). Interindividual variability was low, with CVs. of 1.24% to 5.52% for all doses [9]. aPTT values returned to baseline within 72 h. The maximal effects on aPTT were observed after 4 h and 8 h of administration under fasted and fed states, respectively. Similar kinetic was observed on the dose and time dependent inhibition of FXIa and FXI activities. FXIa activity was measured using a fluorogenic substrate, while FXI (clotting) activity was measured using a modified APTT assay [9]. The maximal decrease in FXIa and FXI activity relative to baseline were 0.01 and 0.52, respectively (Figure 5). These effects were observed approximately 1 h after administration of oral solution and 2–4 h after administration of tablets. The inhibition of FXIa activity was faster in fasted state compared to fed state, with maximal effects observed 4 and 8 h after administration, respectively.

#### 3.2.2. Milvexian

##### Mechanism of Action

Milvexian is a potent small molecule that inhibits the active form of FXIa with high affinity [45]. This molecule is one of the first oral FXIa inhibitors being developed as new potential antithrombotic drug. Milvexian interacts to FXIa with a Ki equal to 0.11 nM and shows an aPTT EC1.5× of 0.50 μM [45]. The molecule exhibited exquisite selectivity (>5000-fold) for 15 of the 17 proteases tested, such as plasma kallikrein (400-fold) and chymotrypsin (300-fold) [45].

##### Pharmacokinetics

A specific PK study with single and multiple ascending doses of milvexian has been recently reported in healthy volunteers (Table 8) [33]. After single oral doses milvexian reached the maximum plasma concentration after 3 h (T_max_) at all doses tasted, ranging from 4 to 500 mg. T_1/2_ ranged from 8.3 to 13.8 h across all doses, and C_max_ or exposure were shown to be proportional with the dose ranging from 20 to 200 mg. A partial saturation of the exposures was observed at doses of 300 mg or higher. Under fed conditions, the T_max_ was very similar than fasting (4 h at doses of 200 and 500 mg), however the elimination rate was faster when milvexian was given in the presence of food (T_1/2_ shortened by 1.5 h). Food increased the bioavailability of milvexian by 1.4-fold for 200-mg and by 2-fold for 500-mg. Milvexian is mainly eliminated by the liver, with only 6.9–17.8% of administered dose found in the urine from 0 to 24 h postdose. These range increased significantly in fed state (16.3–17.8%) probably due to the higher bioavailability.

After multiple ascending doses, a partial accumulation was observed at doses of 200 and 500 mg q.d. (1.27–1.75 for both C_max_ and AUC) and 200 mg b.i.d. (3.99 for AUC and 2.68 for C_max_). Results of this first-in-human study showed milvexian was safe and well-tolerated in healthy volunteers at the single dose up 500 mg and multiple doses up to 200 mg b.i.d. and 500 mg q.d. [33].

##### Pharmacodynamic

Single and multiple ascending doses of milvexian prolonged aPTT in a dose-dependent manner with a maximum prolongation that coincided with T_max_ [33]. An almost two-fold increase of aPTT from baseline was observed at the highest single dose of 500 mg. The magnitude of the reduction of FXIa activity was related to drug exposure, while prothrombin time was not affected by milvexian (Figure 6) [33].

### 3.3. Antisense Oligonucleotides (ASO)

#### 3.3.1. Fesomersen (IONIS-FXI-LRx/ISIS 416858/ BAY2306001)

##### Mechanism of Action

Fesomersen is a human and monkey cross-reactive second-generation 2′-methoxyethoxy (2′-MOE) ASO inhibitor that is currently under clinical development. The mechanism of action of fesomersen is to target FXI mRNA in the liver and preventing factor synthesis and activity. Preclinical study conducted in cynomolgus monkeys showed a liver specific reduction of FXI mRNA expression in a dose- and duration-dependent manner. Fesomersen reduced by 50% and 90% hepatic FXI mRNA expression after 6 weeks of treatment with 12 and 40 mg/kg dose, respectively. The drug was administered with by i.v. infusion, followed by a loading dose and then a s.c. injection every week [46]. The effect on FXI mRNA was associated to a significant reduction of plasma FXI protein activity with a maximal 75% reduction reached by day 21 at the 40 mg/kg dose (Figure 7). At this dose, following cessation of treatment, a complete FXI plasma activity recovery was still evident after 13 weeks of a treatment-free period.

The functional consequence of pharmacologic inhibition of FXI was a significant aPTT prolongation in a dose- and duration-dependent manner following fesomersen treatment. The aPTT increase was observed after 2 weeks of treatment initiation and reached maximal level (+75%) by day 93 at the 40 mg/kg dose (Figure 7). The aPTT returned to basal by the end of the 13-week treatment-free period.

##### Pharmacokinetics

The pharmacology profile of fesomersen has been first reported from a preliminary study conducted in healthy volunteers after single or multiple ascending-dose of s.c. injection at 50, 100, 200 and 300 mg/kg. This analysis indicated that T_1/2_ of fesomersen across doses was approximately equal to 20 days [36]. A more detailed PK study evaluated the PK profile of fesomersen in patients with end-stage renal disease (ESRD) [38]. The exposure of fesomersen, measured during the first 24 h post injection, did not differ between pre and post hemodialysis (AUC 142 h × mg/mL [CV 39.7%] vs. 136 [CV 45.6%] h × mg/mL) [38].

T_max_ values were equal to 5.97 h (range 2.98–10.0) when administered pre-HD and to 6.03 h (range 4.0–9.95) post-HD. In the randomized multiple-dosing study, patients were allocated to the 200 mg and 300 mg dose or placebo. This analysis revealed a stable C_max_ between the first dose on day 1 and after the last dose on day 78 for both the 200 mg and 300 mg doses and a t_1/2_ of 16.9 days (CV 39.9%) for the 200 mg dose and 13.1 days (CV 35.1%) for the 300 mg dose (Table 9). There was negligible urinary excretion of fesomersen (0.0424% dose excreted at 78 days with urine). C_max_ was dose dependent and similar between day 1 and day 78, suggesting there was no accumulation of fesomersen after weekly administrations (Table 9). After the last dose, fesomersen concentrations decreased with time, with a t_1/2_ of approximately 2 weeks.

##### Pharmacodynamics

In the single ascending dose, the administration of 200 and 300 mg of fesomersen significantly reduced FXI activity 1-week after dosing. In the multiple ascending doses, the treatment with fesomersen demonstrated a robust, sustained and dose-dependent reduction in FXI expression and activity (Table 10) [36]. These reductions were accompanied by a concomitant increase in aPTT.

A subsequent study evaluated the PK and pharmacodynamics for both single 300-mg dose and multi-dose of 200 and 300 mg every 12 weeks in 49 patients with end stage renal disease on dialysis (ESRD) [37,38]. The inhibitory effect of fesomersen on FXI activity, observed at day 85, was equal to 56.0% in the 200 mg group and 70.7% in the 300 mg group compared to basal (Figure 8) [38]. Similar reductions were observed in the plasma concentration of FXI antigen (−65.5% for 200 and −78.1% for 300 mg). The reductions in FXI activity levels were maximal at approximately 6 weeks after treatment initiation, consistent with previous studies and reflective of the antisense mechanism coupled with the long half-life of FXI protein (approximately 55 h) [39]. Reduction in FXI activity and antigen levels with fesomersen determined a dose-dependent prolongation of aPTT but no clinically significant changes in PT or INR. The maximum effect on aPTT prolongation was observed on day 92 with 46.1 s at 200 mg and 49.4 s at 300 mg, and 36.0 s in the placebo [38].

Thus, the treatment with fesomersen demonstrated a statistically significant and sustained reduction of FXI activity and expression associated to a prolonged aPTT [38]. These data have supported the clinical development of fesomersen as a novel approach for the treatment and prevention of thromboembolic disorders.

## 4. Completed Phase 2 Trials

As for fully mAbs (Table 2), the ANT-005 TKA study showed that a unique post-operative dose of abelacimab at the dosages of 75 or 150 mg vs. one daily subcutaneous enoxaparin (40 mg) was significantly non-inferior in terms of rates of symptomatic and asymptomatic VTE (4% and 5% vs. 22% with enoxaparin, respectively) in patients undergoing total knee arthroplasty [23]. No bleeding complications were recorded [23]. Similarly, the FOXTROT Clinical Trial randomized 813 patients to receive either single intravenous postoperative osocimab on the day after total knee arthroplasty at 4 dosages (0.3 mg/kg [*n* = 107], 0.6 mg/kg [*n* = 65], 1.2 mg/kg [*n* = 108], 1.8 mg/kg [*n* = 106] or preoperative osocimab at 2 dosages (0.3 mg/kg [*n* = 109], 1.8 mg/kg [*n* = 108]) vs. subcutaneous enoxaparin (40 mg) once daily (*n* = 105) or 2.5 mg of oral apixaban twice daily (*n* = 105) for at least 10 days after knee arthroplasty [25]. Only preoperative osocimab at 1.8 mg/kg was found to be superior to enoxaparin (risk difference 15.1% [90% CI 4.9–25.2%]) for the outcome symptomatic or asymptomatic VTE, while no regimen of osocimab was found superior to apixaban [25]. As for bleeding, major or clinically relevant nonmajor bleeding was observed in 4.7% of patients receiving osocimab, 5.9% receiving enoxaparin, and 2% receiving apixaban [25]. Finally, in a very recent randomized, double-blind, phase 2 study, 24 patients with end stage renal disease (ESRD) undergoing heparin-free hemodialysis were randomized to receive a single pre-dialysis dose of AB023 (0.25 or 0.5 mg/kg) or placebo in a 2:1 ratio. Occlusive events requiring hemodialysis circuit exchange were less frequent and levels of thrombin-antithrombin complexes lower after AB023 administration vs. placebo [28].

As for small molecules, in the PACIFIC-AF 755 patients aged 45 years or older with AF, a CHA2DS2-VASc score ≥ 2 (male) or ≥ 3 (female), and increased bleeding risk were randomized to asundexian at the dosages of 20 mg or 50 mg once daily or apixaban 5 mg twice daily [30]. The primary endpoint was the composite of major or clinically relevant non-major bleeding. Incidence ratio for this endpoint for pooled asundexian (four events) versus apixaban (six events) was 0.33 (0.09–0.97). The rate of any adverse event was similar in the three treatment groups: 118 (47%) with asundexian 20 mg, 120 (47%) with asundexian 50 mg, and 122 (49%) with apixaban. The Authors concluded that asundexian at both doses resulted in lower rates of bleeding compared with standard dosing of apixaban [30]. In a very recent multicenter trial, 1601 patients with recent (within 5 days) acute myocardial infarction (MI) were randomized to oral asundexian at the dosages of 10, 20, or 50 mg or placebo once daily for 6–2 months. Patients received also dual antiplatelet therapy with aspirin plus a P2Y12 inhibitor [31]. The main safety outcome was bleeding, while the efficacy outcome was a composite of cardiovascular death, MI, stroke, or stent thrombosis comparing pooled asundexian 20 and 50 mg doses vs. placebo. Over a median follow-up of 368 days, the safety outcome occurred in 30 (7.6%), 32 (8.1%), 42 (10.5%), and 36 (9.0%) patients receiving asundexian 10, 20, 50 mg, and placebo (pooled asundexian vs. placebo: HR 0.98, 90% CI 0.71–1.35). The efficacy outcome occurred in 27 (6.8%), 24 (6.0%), 22 (5.5%), and 22 (5.5%) patients assigned asundexian 10, 20, 50 mg, and placebo (pooled asundexian 20 and 50 mg vs. placebo: HR 1.05, 90% CI 0.69–1.61). Asundexian caused a dose-related inhibition of FXIa activity with 50 mg resulting in >90% inhibition. The Authors concluded that 3 doses of asundexian, when added to aspirin plus a P2Y12 inhibitor, resulted in dose-dependent, near-complete inhibition of FXIa activity without a significant increase in bleeding and a low rate of ischemic events at 4 weeks, thus supporting the investigation of asundexian 50 mg daily in an adequately powered clinical trial [31].

The AXIOMATIC-TKR study randomized 1242 patients undergoing total knee arthroplasty to receive one of seven postoperative dosages of milvexian (25 mg, 50 mg, 100 mg, or 200 mg twice daily or 25 mg, 50 mg, or 200 mg once daily) or enoxaparin (40 mg once daily) [34]. In the milvexian twice daily arms, symptomatic or asymptomatic VTE developed in 27 of 129 (21%) taking 25 mg, in 14 of 124 (11%) taking 50 mg, in 12 of 134 (9%) taking 100 mg, and in 10 of 131 (8%) taking 200 mg. In patients who received milvexian once daily, VTE developed in 7 of 28 (25%) taking 25 mg, in 30 of 127 (24%) taking 50 mg, and in 8 of 123 (7%) taking 200 mg, vs. 54 of 252 patients (21%) taking enoxaparin. No differences were found in terms of bleeding (4% in patients receiving milvexian and 4% in those taking enoxaparin). Thus, oral milvexian was effective for the prevention of VTE, particularly with twice-daily regimen, and associated with a low risk of bleeding [34]. Very recently, the results of the AXIOMATIC-SPP trial were presented as late breaking research at the ESC Congress 2022 [35]. The study randomized 2366 patients with a mild-to-moderate acute non-lacunar ischemic stroke (National Institutes of Health Stroke Scale score ≤7) or a high-risk transient ischemic attack (TIA, ABCD^2^ score ≥6) with evidence of arterial atherosclerosis to receive one of five milvexian dosages (25, 50, 100, 200 mg twice daily, 25 mg once daily) or placebo daily for 90 days. All participants received open-label aspirin and clopidogrel for 21 days, followed by open-label aspirin from day 22 to 90 days. The primary efficacy endpoint was a composite of ischemic stroke during treatment or incident infarct on brain MRI at 90 days, while the main safety endpoint was major bleeding. Milvexian resulted in a reduction of ischemic stroke risk (namely an approximately 30% relative risk reduction) at all dosages except 200 mg twice daily vs. palcebo. The incidence of major bleeding was low overall and for milvexian 25 mg once daily and twice daily was similar to placebo, while a moderate increase was observed in the milvexian dose 50 mg twice daily and above (the majority of which were gastrointestinal bleeds), with no apparent dose-response. No fatal bleeding was detected [35].

As for ASO, the FXI-ASO TKA study randomized 300 patients undergoing total knee arthroplasty to receive fesomersen (IONIS-FXI-L_RX_) at the dosages of 200 mg or 300 mg or 40 mg of enoxaparin (40 mg) once daily [39]. FXI-ASO at 200 mg was non-inferior (27%), while at 300 mg was superior (4%) to enoxaparin (30%) (*p* < 0.001) for the outcome symptomatic or asymptomatic VTE prevention. Interestingly, rate of bleeding was inferior in the FXI-ASO groups (both 3%) than in the enoxaparin group (8%) [39]. In patients with ESRD requiring hemodialysis, 43 subjects were randomized to 200 mg or 300 mg IONIS-FXI-L_RX_ or placebo for 12 weeks, showing that major bleeding events occurred in 0 participants taking 200 mg, 1 (6.7%) participant taking 300 mg, and 1 (7.7%) taking placebo; bleeding episodes were not considered associated with the treatment [38].

Ongoing phase III clinical study with anti-FXI compounds are reported in Table 11.

## 5. Discussion and Conclusions

The current oral anticoagulants act by reducing plasma levels of prothrombin and FX (VKAs) or inhibiting their activated forms (DOACs). As a matter of fact, thrombin and FXa are important in the coagulation initiation process as they are in coagulation propagation and thrombosis development, thus their inhibition compromise hemostasis likely leading to bleeding events [11]. The hemorrhagic risk puts a limitation on the anticoagulation intensity that can be administered, with some patients with further increased bleeding risk not eligible for therapy. FXI, on the contrary, has an ancillary role in the coagulation initiation process, while its relative contribution to coagulation propagation and thrombotic process appears to be greater [11,12]. Considering also that patients with severe FXI deficiency rarely have spontaneous hemorrhage and are not at increased risk for bleeding into the central nervous system or gastrointestinal tract, drugs targeting FXI or FXIa should be associated with a relatively low risk for serious bleeding complications [12,13]. Different pharmacological approaches are currently in clinical development to inhibit FXIa and pharmacodynamic studies clearly showed an effective inhibition of this coagulation factor. However, the laboratory results (e.g., aPTT, FXI activity, ROTEM) cannot be considered a valid predictor of positive clinical outcomes. Indeed, extensive phase 2 and 3 clinical trials are underway to study the clinical efficacy of these approaches.

In phase 2 studies inhibition of FXI or FXIa has proven to be effective at preventing VTE in patients undergoing total knee arthroplasty, or for prevention of major adverse vascular events in patients with end-stage kidney disease undergoing hemodialysis or as adjuncts to antiplatelet therapy for prevention of recurrent ischemic events in patients with acute myocardial infarction or non-cardioembolic stroke. Finally, the head to head comparison between FXI inhibitors and a DOAC (apixaban) for stroke prevention in patients with atrial fibrillation has recently confirmed a more favorable profile in term of safety for the former (42). Ongoing phase 3 clinical trials will confirm the safety and efficacy of these very promising compounds in clinical peculiar situations such as cancer associated thrombosis and high-risk atrial fibrillation (Table 11). Should the efficacy of FXI inhibitors as anticoagulant without impairing the hemostasis be proven in these settings, it would provide an innovative therapeutic option characterized by high manageability, longer half-life, renal tolerability, and fewer drug interactions. 

## Figures and Tables

**Figure 1 jcm-11-06314-f001:**
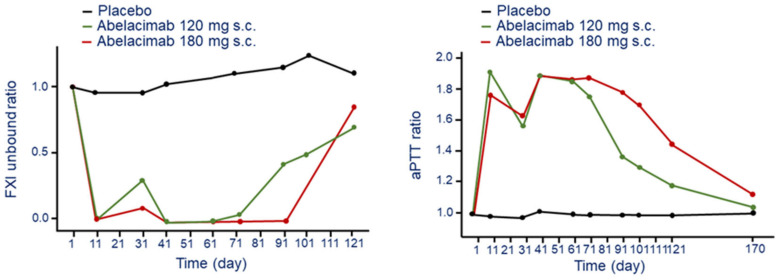
Time and dose-dependent effect of abelacimab on bound FXI (**left panel**) and aPTT (**right panel**) following monthly s.c. administration. Ratios were calculated by dividing by the subject’s baseline values. Modified from Yi A.B. et al. [22]. F: factor; aPTT: activated partial thromboplastin time.

**Figure 2 jcm-11-06314-f002:**
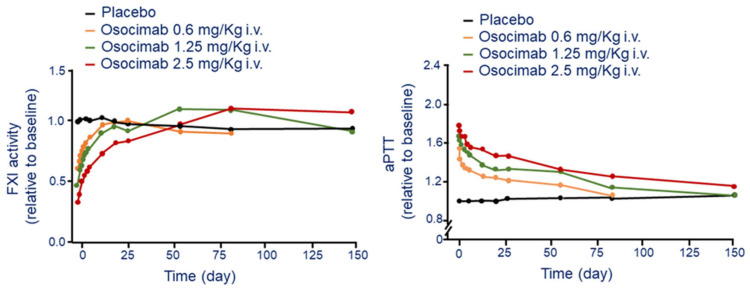
Time and dose-dependent effect of osocimab on FXI activity (**left panel**) and aPTT (**right panel**) following a single i.v. administration. Modified from Thomas D et al. [24]. F: factor; aPTT: activated partial thromboplastin time.

**Figure 3 jcm-11-06314-f003:**
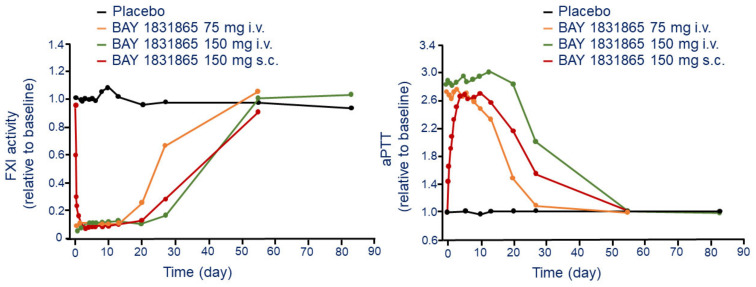
Time and dose-dependent effect of BAY 1831865 on FXI activity (**left panel**) and kaolin-induced aPTT (**right panel**) following a single i.v. and s.c. administration. Modified from Nowotny B. et al. [26]. F: factor; aPTT: activated partial thromboplastin time.

**Figure 4 jcm-11-06314-f004:**
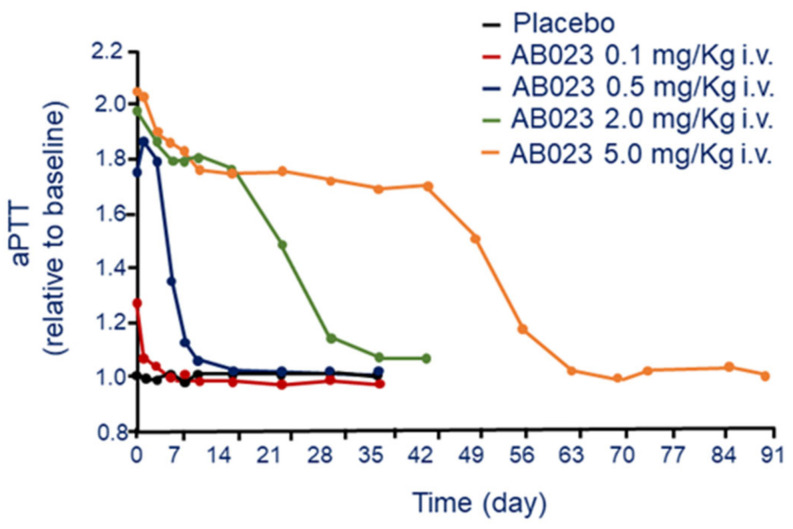
Dose-dependent effect of AB 023 on aPTT following a single i.v. administration. Modified from Lorentz, C.U. et al. [27]. aPTT: activated partial thromboplastin time.

**Figure 5 jcm-11-06314-f005:**
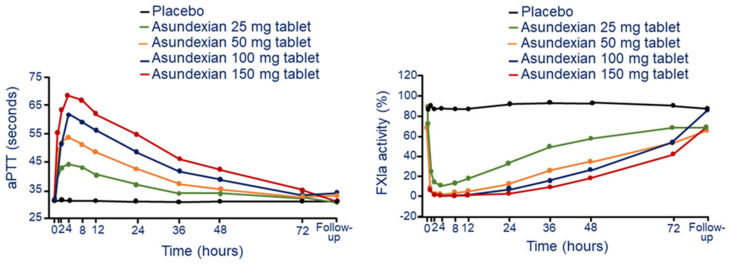
Dose and time-dependent effect of asundexian on aPTT (**left panel**) and FXIa activity (**right panel**) following a single oral tablet administration. Modified from Thomas, D. et al. [29]. F: factor; aPTT: activated partial thromboplastin time.

**Figure 6 jcm-11-06314-f006:**
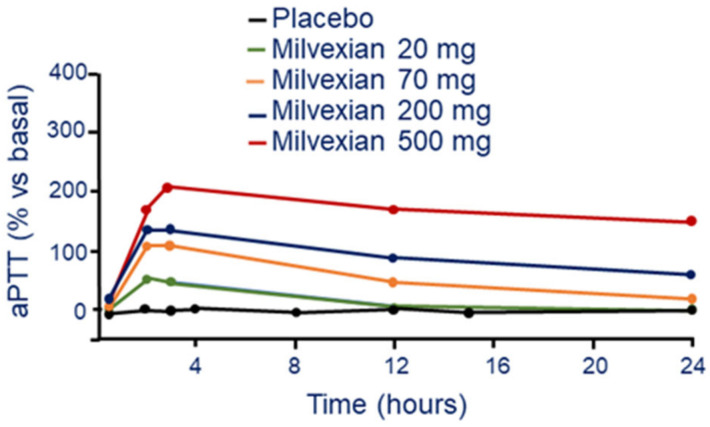
Dose and time-dependent effect of milvexian on aPTT following a single oral tablet administration. Modified from Perera, V et al. [33]. aPTT: activated partial thromboplastin time.

**Figure 7 jcm-11-06314-f007:**
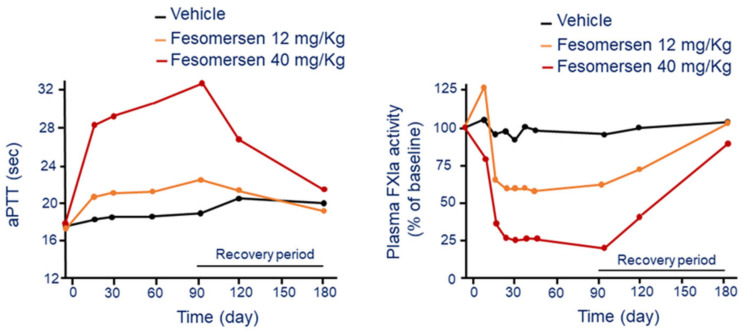
Dose and time-dependent effect of fesomersen on aPTT (**left panel**) and FXIa activity (**right panel**). Modified from Younis, H.S. et al. [46]. F: factor; aPTT: activated partial thromboplastin time.

**Figure 8 jcm-11-06314-f008:**
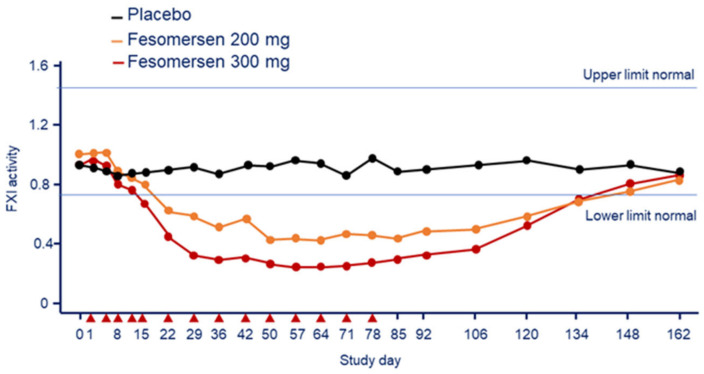
Effect of fesomersen 200 mg and 300 mg on FXI activity compared with placebo during the 78-day treatment period and subsequent 85-day washout period. Red arrowheads indicate dosing day. Modified from Walsh, M. et al. [38]. F: factor.

**Table 1 jcm-11-06314-t001:** Unresolved Issues in Established Indications for oral anticoagulation with potential indication for antiFXI.

Medical Condition	Unresolved Issues
Atrial fibrillation	End-stage renal disease (ClCR < 15 mL/min) Elderly patients (age ≥ 85 years) Concomitant acute coronary syndrome Concomitant heart valvular prothesis Medical devices Resumption of anticoagulation after major bleeding
Venous thromboembolism	End-stage renal disease Patients with gastro-intestinal or genito-urinary cancer Drug-drug interactions Antiphospholipid antibody syndrome

ClCR: creatinine clearence.

**Table 2 jcm-11-06314-t002:** FXI inhibitors currently under clinical development for the treatment of TEV and AF.

Compound Name	Developer	Target	Inhibitory Class	Phase of Development	Population	Trial Design	Number of Patients	Ref.
Abelacimab (MAA868)	Anthos Therapeutics Novartis AG	FXI and FXIa	Fully human mAb IgG1	Phase 1	Healthy volunteers and patents with AF	Randomized, subject-and investigator-blinded, placebo-controlled study	18	[22]
				Phase 2	Patients undergoing total knee arthroplasty	Open-label, parallel-group trial	412	[23]
Osocimab (BAY1213790)	Bayer Pharmaceuticals	FXIa	Fully human mAb IgG1	Phase 1	Healthy volunteers	Randomized, single-blind, placebo-controlled, dose-escalation study	81	[24]
				Phase 2	Patients undergoing total knee arthroplasty	Randomized, open-label, adjudicator-blinded, noninferiority trial	813	[25]
BAY1831865	Bayer Pharmaceuticals	FXI	Humanized mAb IgG1	Phase 1	Healthy volunteers	Randomized, single-blind (participant), parallel-group, placebo-controlled, dose-escalation	70	[26]
AB023/ Xisomab 3G3	Aronora	FXIa	Fully human monoclonal IgG1 antibody	Phase 1	Healthy volunteers	Randomized, double-blind, placebo-controlled, single ascending bolus dose study	21	[27]
				Phase 2	Patients with end stage renal disease on chronic hemodialysis	Randomized, Double-Blind, Placebo-Controlled Study	24	[28]
Asundexian (BAY 2433334)	Bayer Pharmaceuticals	FXIa	Small molecule	Phase 1	Healthy volunteers	Single-blind, placebo-controlled, dose-escalation study	70	[29]
				Phase 2	Patients with AF	Randomized, double-blind, double-dummy, dose-finding	755	[30]
				Phase 2	Patients with	Randomized, placebo-controlled, double-blind, parallel-group, dose-finding	1601	[31]
Milvexian (BMS-986177 JNJ- 70033093)	Janssen of Johnson and Johnson and Bristol-Myers Squibb	FXIa	Small molecule	Phase 1	Healthy and mild or moderate hepatic impairment	Open-label	26	[32]
				Phase 1	Healthy volunteers	Randomized, double-blind, placebo-controlled, single and multiple ascending doses	94	[33]
				Phase 2	Patients undergoing total knee arthroplasty	Randomized, parallel-group	1242	[34]
				Phase 2	Patients with ischemic stroke	Randomized, double-Blind, placebo-controlled, dose-Ranging Study	2366	[35]
Fesomersen (IONIS-FXI-L_RX_)	IONIS and Bayer	FXIa	Antisense Oligonucleotide	Phase 1	Healthy volunteers	Placebo-controlled, dose escalation study	36	[36]
				Phase 1	End-Stage Renal Disease on Hemodialysis patients	Open-label single-dose and double-blind multiple doses	49	[37]
				Phase 2	Patients with end stage renal disease on chronic hemodialysis	Double-blind, randomized	43	[38]
				Phase 2	Patients undergoing total knee arthroplasty	open-label, parallel-group	300	[39]

AF: atrial fibrillation; F: factor; Ig: Immunoglobulin.

**Table 3 jcm-11-06314-t003:** Plasma pharmacokinetic parameters of abelacimab. Geometric Mean (Geometric Mean CV%) abelacimab PK parameters. For T_max_, data are presented as median (min-max). Data are from Koch A.W. et al. [40] and Yi B.A. et al. [22]. C_max_: Maximum plasma concentration; T_max_: time to reach C_max_. AUC: area under the curve; T_1/2_: half-life time; Vd: volume of distribution; CL: clearance. BMI: body mass index.

Parameter	Abelacimab 150 mg s.c. (*n* = 8)	Abelacimab 150 mg i.v. (*n* = 6)	Abelacimab 150 mg i.v. (*n* = 6) BMI > 35
C_max_ (µg/mL)	11.6 (28.8)	52.3 (17.7)	36.7 (16.8)
T_max_ (h)	7.00 (7.00–14.0)	2.00 (1.50–3.03)	1.75 (1.20–2.02)
AUC (h × µg/mL)	9696 (15.4)	21,782 (18.3)	12,543 (26.6)
T_1/2_ (h)	580.8 (16.2)	595 (29.7)	621 (17.1)
Vd (L)	13.3 (24.2)	5.00 (24.1)	8.29 (20.8)
CL (L/h)	0.0158 (16.7)	0.00653 (17.6)	0.0120 (26.6)

**Table 4 jcm-11-06314-t004:** Plasma pharmacokinetic parameters of osocimab. Geometric Mean (Geometric Mean CV%) osocimab PK parameters. For T_max_, data are presented as median (min-max). Data are from Thomas D. et al. [24]. C_max_: Maximum plasma concentration; T_max_: time to reach C_max_. AUC: area under the curve; T_1/2_: half-life time; Vd: volume of distribution; CL: clearance.

Parameter	Osocimab 0.3 mg/Kg i.v. (*n* = 6)	Osocimab 0.6 mg/Kg i.v. (*n* = 8)	Osocimab 1.25 mg/Kg i.v. (*n* = 8)	Osocimab 2.5 mg/Kg i.v. (*n* = 8)
C_max_ (mg/L)	8.01 (11.6)	22.3 (14.2)	42.6 (12.5)	78.5 (13)
T_max_ (h)	1.52 (1.02–2.03)	2.02 (1.07–12.1)	4 (1.02–8)	3 (1–8)
AUC (h × µg/mL)	3610 (17.9)	7200 (12.7)	14,200 (15.8)	23,700 (16)
T_1/2_ (h)	748 (14.3)	729 (12.7)	809 (11.8)	1050 (16.5)
Vd (L)	5.6 (9.25)	5.43 (12.1)	6.58 (17.2)	8.69 (14.7)
CL (L/h)	0.00535 (9.2)	0.00565 (12.8)	0.00663 (19.3)	0.00787 (18.1)

**Table 5 jcm-11-06314-t005:** Plasma pharmacokinetic parameters of BAY 1831865. Geometric Mean (Geometric Mean CV%) BAY 1831865 PK parameters. For T_max_, data are presented as median (min-max). Data are from Nowotny B. et al. [26]. C_max_: Maximum plasma concentration; T_max_: time to reach C_max_. AUC: area under the curve; AUC/D: ratio of AUC and dose; T_1/2_: half-life time; Vd: volume of distribution; CL: clearance.

Parameter	BAY 1831865 150 mg i.v.	BAY 1831865 150 mg s.c
C_max_ (mg/L)	48.3 (9.47)	10.9 (67.4)
T_max_ (h)	1.94 (1.00–3.95)	96.0 (48–239)
AUC (h × µg/mL)	10.900 (16.7)	5160 (61.1)
AUC/D (h /L)	72.9 (16.7)	34.4 (61.1)
T_1/2_ (h)	208 (19.8)	217 (36.5)
Vd (L)	0.0137 (16.7)	0.0291 (61.1)
CL (L/h)	0.0137 (16.7)	0.0291 (61.1)

**Table 6 jcm-11-06314-t006:** Plasma pharmacokinetic parameters of AB023. Geometric Mean (Geometric Mean CV%) AB023 PK parameters. For T_max_, data are presented as median (min-max). Data are from Lorentz, C.U. et al. [27]. C_max_: Maximum plasma concentration; T_max_: time to reach C_max_. AUC: area under the curve; T_1/2_: half-life time; Vd: volume of distribution; CL: clearance.

Parameter	AB023 0.1 mg/Kg i.v.	AB023 0.5 mg/Kg i.v.	AB023 2.0 mg/Kg i.v.	AB023 5.0 mg/Kg i.v.
C_max_ (µg/mL)	122.7 (23.3)	11210 (9.1)	42,510 (12.3)	127,200 (2.6)
T_max_ (h)	0.084 (0.08–0.09)	0.649 (0.26–3.02)	0.088 (0.08–0.25)	0.387 (0.26–3.00)
AUC (h·ng/mL)	57.0 (46.5)	361,800 (21.9)	5,540,000 (23.9)	28,120,000 (11.8)
T_1/2_ (h)	1.33	16.64 (9.4)	60.63 (7.3)	121.49 (33.4)
Vd (L)	69.11	2.52 (23.4)	3.72 (11.3)	4.31 (17.4)
Cl (L/h)	37.46	0.094 (19.1)	0.026 (11.5)	0.014 (21.4)

**Table 7 jcm-11-06314-t007:** Plasma pharmacokinetic parameters of asundexian. Geometric Mean (Geometric Mean CV%) asundexian PK parameters. For T_max_, data are presented as median (min-max). Data are from Thomas, D et al. [29]. C_max_: Maximum plasma concentration; T_max_: time to reach C_max_. AUC: area under the curve; T_1/2_: half-life time.

Parameter	Asundexian Oral Solution			Asundexian Tablets		
	5 mg	12.5 mg	25 mg	25 mg	50 mg	100 mg
C_max_ (μg/L)	91.4 (15.9)	205 (19.1)	372 (13.5)	320 (18.3)	617 (15.6)	1230 (25.9)
AUC (h·ng/mL)	1570 (18.8)	3770 (24.9)	6630 (25.4)	5940 (19.9) 13	200 (16.1)	27 600 (23.5)
T_max_ (h)	1.0 (0.75–1.50)	1.0 (0.75–1.50)	1.0 (0.75–1.50)	2.5 (1.0–6.0)	3.0 (1.0–4.0)	3.0 (0.8–6.0)
T_1/2_ (h)	14.5 (15.7)	16.0 (18.1)	15.2 (21.6)	14.2 (20.0)	17.4 (12.4)	16.3 (13.7)

**Table 8 jcm-11-06314-t008:** Plasma pharmacokinetic parameters of milvexian. Geometric Mean (Geometric Mean CV%) milvexian PK parameters. For T_max_, data are presented as median (min-max). Data are from Perera, V et al. [33]. C_max_: Maximum plasma concentration; T_max_: time to reach C_max_. AUC: area under the curve; T_1/2_: half-life time.

Parameter	Milvexian 20 mg Fasted	Milvexian 60 mg Fasted	Milvexian 200 mg Fasted	Milvexian 200 mg Fed	Milvexian 300 mg Fasted	Milvexian 500 mg Fasted	Milvexian 500 mg Fed
C_max_ (ng/mL)	126 (36.7)	337 (23.6)	1068 (98.6)	1639 (20.9)	1017 (34.1)	1853 (32.0)	3359 (39.6)
AUC (ng·h/mL)	1220 (20.3)	3793 (13.7)	12,471 (68.0)	17,811 (19.4)	14,588 (29.9)	20,991 (26.8)	44,330 (25.0)
T_max_, (h)	3.0 (1.0–4.0)	3.0 (2.0–4.0)	3.0 (2.0–4.0)	4.0 (3.0–6.0)	3.0 (2.0–4.0)	3.0 (2.0–4.0)	4.0 (4.0–6.0)
T_1/2_ (h)	8.3 (17.0)	9.9 (13.7)	10.5 (28.8)	9.0 (16.1)	13.8 (21.0)	12.2 (14.4)	10.7 (12.1)

**Table 9 jcm-11-06314-t009:** PK parameters of fesomersen after repeated doses of 200 mg and 300 mg twice weekly for 15 days then weekly for 78 days. Data are from Walsh, M. et al. [38]. Data are presented as geometric mean (geometric % CV). NA: not available. C_max_: Maximum plasma concentration; T_1/2_: half-life time.

Parameter	200 mg/Kg Fesomersen		300 mg/Kg Fesomersen	
	(Day 1)	(Day 78)	(Day 1)	(Day 78)
C_max_ (μg/mL)	8.24 (42.0)	10.3 (36.8)	14.3 (46.5)	12.3 (41.1)
T_1/2_ (h)	NA	16.9 (39.9)	NA	13.1 (35.1)

**Table 10 jcm-11-06314-t010:** Mean (%) change from baseline in the multiple ascending dose regimen of fesomersen. Data are from Liu, Q. et al. [36].

	FXI Antigen Reduction	FXI Activity Reduction	aPTT Prolongation
Placebo (pooled, *n* = 9)	3	3	1
Fesomersen 50 mg (*n* = 9)	31 (*p* = 0.0013)	15 (*p* = 0.1)	8 (*p* = 0.14)
Fesomersen 100 mg (*n* = 9)	54 (*p* < 0.0001)	45 (*p* < 0.0001)	18 (*p* = 0.0078)
Fesomersen 200 mg (*n* = 9)	78 (*p* < 0.0001)	71 (*p* < 0.0001)	67 (*p* < 0.0001)

**Table 11 jcm-11-06314-t011:** Phase III ongoing trials assessing anti-FXI drugs in the prevention of thrombosis.

Compound Name	Inhibitory Class	Setting	Trial ID	Regimen	Comparator	Sample Size	Status
Abelacimab (MAA868)	Fully human mAb IgG1	Cancer-associated thrombosis	NCT05171075 (Magnolia)	IV/SC	Dalteparin	1020	Recruiting
Abelacimab (MAA868)	Fully human mAb IgG1	Cancer-associated thrombosis	NCT05171049 (Aster)	IV/SC	Apixaban	1655	Recruiting
Asundexian (BAY 2433334)	Small molecule	Atrial fibrillation at risk for stroke	OCEANIC-AF	Oral	Apixaban	~2000	Upcoming
Asundexian (BAY 2433334)	Small molecule	Non-cardioembolic ischemic stroke or high-risk ischemic attack	OCEANIC-Stroke	Oral	placebo-controlled study on top of standard-of-care antiplatelet therapy	~2000	Upcoming

## Data Availability

Not applicable.

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
