# Peer review of "Clinical Pharmacology of Factor XI Inhibitors: New Therapeutic Approaches for Prevention of Venous and Arterial Thrombotic Disorders"

_jcm, 2022, doi:10.3390/jcm11216314_

Round 1
Reviewer 1 Report
E Campello et al. provide a comprehensive and update review of clinical development of FXI inhibitors. We should thank the authors for this very useful work.
I have no major remarks, but just comments:
The mechanisms of inhibition of FXI/XIa, according to the site of interaction with the target, are different. I regret that a cartoon does not illustrate this. Clearly, this class of new drugs is more heterogeneous than, for example, FXa inhibitors. As a result, a drug could demonstrate preferential benefits in one indication and not in others.
It should be emphasized that in most of completed phase 2 clinical trials, FX/XIa inhibitors are compared with current anticoagulant strategies (direct anticoagulant or heparins). The results strongly suggest the expected safety and efficacy of FXI/XIa inhibitors but generally fail to demonstrate a significant reduction of thrombotic occurrence, probably due to the small size of these trials. The benefit/risk of these new drugs remains to be confirmed in phase 3 large studies.
Should the benefit of FXI/XIa inhibitors over current anticoagulants be only marginal, a major interest would be for patients difficult to manage with current anticoagulants (strong drug interactions, severe liver or renal disorders, contraindication to heparin, etc.) or to prevent vascular thrombosis in patients with critically conditions where contact activation plays a role (sepsis, infection such as Covid 19, inflammatory diseases ...).
Minor comments
Page 12, lines 348-349, the sentence is sized.
Page 15, line 419, “of magnitude” is repeated
Page 18, line 525, put a comma after PACIFICA-AF
Author Response
Authors’ response:
Reviewer #1
E Campello et al. provide a comprehensive and update review of clinical development of FXI inhibitors. We should thank the authors for this very useful work.
We thank the Reviewer for the kind comment.
- The mechanisms of inhibition of FXI/XIa, according to the site of interaction with the target, are different. I regret that a cartoon does not illustrate this. Clearly, this class of new drugs is more heterogeneous than, for example, FXa inhibitors. As a result, a drug could demonstrate preferential benefits in one indication and not in others.
We thank the Reviewer for this insightful comment. A comment has been added dealing with the heterogenous mechanisms of action of this drug class and the possible clinical implication (see pg. 19 “Additionally, the final anticoagulation mechanism of anti-FXI/FXIa ihibitors is different according to their different site of interaction. Thus, this class of new drugs is more heterogeneous than DOACs. As a result, the different compounds could demonstrate preferential benefits in one clinical scenario compared to others”).
- It should be emphasized that in most of completed phase 2 clinical trials, FX/XIa inhibitors are compared with current anticoagulant strategies (direct anticoagulant or heparins). The results strongly suggest the expected safety and efficacy of FXI/XIa inhibitors but generally fail to demonstrate a significant reduction of thrombotic occurrence, probably due to the small size of these trials. The benefit/risk of these new drugs remains to be confirmed in phase 3 large studies. Should the benefit of FXI/XIa inhibitors over current anticoagulants be only marginal, a major interest would be for patients difficult to manage with current anticoagulants (strong drug interactions, severe liver or renal disorders, contraindication to heparin, etc.) or to prevent vascular thrombosis in patients with critically conditions where contact activation plays a role (sepsis, infection such as Covid 19, inflammatory diseases ...).
We completely agree with the Reviewer point. A sentence to emphasize this issue has been added at the end of paragraph 4 (see pg 20: “It is worth mentioning that in most of the completed phase 2 clinical trials, FX/XIa inhibitors are compared with current anticoagulant strategies (i.e. low molecular weight heparin or apixaban). The results strongly suggest favourable safety and efficacy profile for of FXI/XIa inhibitors but generally fail to demonstrate a significant reduction of thrombotic occurrence, probably due to the small size of these trials that were not powered for the efficacy endpoint. The benefit/risk ratio of these new drugs remains to be confirmed in phase III large studies. Should the benefit of FXI/XIa inhibitors over current anticoagulants be only marginal, a major interest would be for patients difficult to manage with current anticoagulants (e.g. drug-drug interactions, severe liver or renal disorders, high bleeding risk) or to prevent vascular thrombosis in patients with critically conditions where contact activation plays a role (e.g. central vein catheters, sepsis”).
Minor comments:
Page 12, lines 348-349, the sentence is sized.
Page 15, line 419, “of magnitude” is repeated
Page 18, line 525, put a comma after PACIFICA-AF
The sentence was shortened and typos corrected. Thank you for noticing them.
Reviewer 2 Report
The authors have submitted a review detailing the pharmacology of FXI inhibitors and data from several clinical trials. This paper was very challenging to review due to the piecemeal presentation of data from multiple sources without clear delineation of source and consistency between inhibitor descriptions. I’ve tried to detail edits and suggestions to help clarify but the raw data presented in this manuscript is overwhelming, please look for opportunities to summarize.
General recommendations:
Ideally all figures shown for all the FXI inhibitors should depict the same parameters for easy comparison. Recommend using FXI activity and aPTT. The axes of the graphs should be the same when describing the same result, example Kaolin-induced aPTT can just be labeled aPTT.
When referencing “clotting time” state the method used to analyze, e.g. ROTEM, TEG, aPTT.
Specific recommendations:
After introduction but before discussing the targeting of FXI – consider adding a general description of FXI. Describe structure and function of the factor and include its role in secondary hemostasis. What activates/inhibits, synthesis, etc. Much of this information is interspersed throughout the manuscript and can easily be consolidated into one section for clarity.
Abelacimab, Mechanisms of action – please condense the detailed description of antibody development. Example – “In 2019 a mAB was developed, using a phage display library, that was capable of binding both FXI and FXIa. The Fab portion exhibited high binding affinity…”
Osocimab section, Figure 3, left panel – the figure copied from reference 24 does not represent changes in PTT, rather it correlates to ROTEM results. Please review and correct the pharmacokinetics section for Osocimab. The summarized data does not completely match the source data from ref 26. Eg. Summary statistics only available for 8 doses, the drug was essentially undetectable at <0.06 not 0.15 mg/kg. If you include the 2 lower doses (both n=7) that will make figure 4 very messy. Consider selective doses – 0.15, 1.25 and 10 mg/kg – that span the ranges of analyzed doses. If these doses are selected, please update figure 4.
Bay 1831865 section. When describing the dose dependent inhibition of BAY 1831865 there is a description of the dose 3.5 mg but not 75 mg which is shown in figure 5. Is there a reason for including 75 mg in the figure although it is not specifically called out in the description? Please review and the last paragraph on page 10. The first couple of sentences refer to FXI and FXIa activities. I believe the intent was to refer to aPTT and FXIa?? Also, when referencing “clotting time” (throughout the manuscript, as well as in this paragraph) state the method used to analyze, e.g. ROTEM, TEG, aPTT. The last sentence needs to be clarified with the dose.
In the Asundexian section, clearly state that figure 7 is from in vitro spiking studies, not from in vivo studies. The ref on line 344 should be 32. In the pharmacokinetics section, if reporting both FXIa and FXI results then need to describe the difference between FXIa and FXI activity measurements. It’s not clear from the language that one refers to a proprietary testing method and the other to a clinical test method.
In the Milvexian section, pages 14-15, there is reference to the high selectivity for FXIa (line 387) but the last sentence in the MOA paragraph states that the molecule exhibited >5000-fold selectivity for 15 of the 17 proteases tested. This is contradictory and there is no mention of other tested proteases in the paragraph. Please clarify. Please expand pharmacodynamic section to include FXIa inhibition description and figure. The last sentence of the paragraph (line 419) needs to be corrected to remove repetition (magnitude) and clarified to define “exposure”.
In the ASO section, pages 15-18, Line 423 – sentence states that “Feromersen reduced by 50% and 90% hepatic FXI mRNA after 6 weeks of treatment (figure 10)” what are the 2 different reduction % referring to? Presumably 2 doses that have not been defined in the paragraph?? This data is not shown in figure 10. Line 433 – change to read, “…followed by every other day s.c. loading dose regimen and weekly s.c. maintenance dosing.” Please reword/clarify the sentence starting on line 436, “At this dose, following cessation…” Please confirm that the data shown in figure 10 is adapted from ref 35, figures and doses do not seem consistent with source data. The pharmacodynamics section needs to be edited for clarity. The data in the ASO section is combined from multiple sources without clear delineation which causes inconsistencies. Multiple doses are mentioned from differing studies yet are referenced back to the same figure 10. There is reference to figure 12 (line 486), I believe this should be corrected to figure 11.
In the completed phase 2 trials section, look for opportunities to summarize this data dense section.
Conclusion should include a statement that the association between the laboratory results (e.g. aPTT, FXI activity, ROTEM) and clinical outcomes have not been established.
Tables – general comments. Please define all abbreviations for every table and use consistent reporting of PK parameters, eg t1/2 for table 6 is different from the other tables.
Table 1 – Random capitalization of words and the title is not clear. Consider something like “Challenges of oral anticoagulation that may not be relevant in the setting of targeted FXI anticoagulation.”
Table 2 – Please review for accuracy! Example Abelacimab phase 1 healthy donors (ANT-003) and patients with AF (ANT-004), reference 23. There were 32 healthy subjects enrolled in ANT-003 and 18 subjects in ANT-004. If the intent is to only include those in the AF group then please update the population column to remove healthy volunteers.
Asudexian, Phase 2, missing population (recent MI). Fesomersen, phase 1, change population description to be consistent with rest of the table “Patients with end stage renal disease on chronic hemodialysis”
Table 9 – the parameter row is too busy, edit so that the dose is mentioned once for Day 1 and Day 78.
Other edits for clarity/accuracy.
Page 2, Line 69 – change to “…seen in hemophilia A or B are not typically found in patients with FXI deficiency.”
Page 5, Line 109 – change to “…high binding affinity to FXI and FXIa and inhibited 50% of FXIa activity at…” remove “the”.
Page 5, line 117 – remove “By meaning of this activity”, and change “timer” to “time”.
Page 6, line 124 – Remove “These” from beginning of the sentence.
Page 9, line 253 – should read (table 5).
Page 10, line 262 – Change to “…increase of aPTT compared to baseline (figure 5).”
Page 10, paragraph starting on line 272. This paragraph consists of 2 sentences that state the same findings. Select one.
Page 11, AB023 section does not include ref 29, only ref 30 regarding neonatal Fc receptor.
Page 12, pharmacodynamics paragraph – combine the 2nd and 3rd sentences to eliminate repetition.
Author Response
Reviewer #2
The authors have submitted a review detailing the pharmacology of FXI inhibitors and data from several clinical trials. This paper was very challenging to review due to the piecemeal presentation of data from multiple sources without clear delineation of source and consistency between inhibitor descriptions. I’ve tried to detail edits and suggestions to help clarify but the raw data presented in this manuscript is overwhelming, please look for opportunities to summarize.
We thank the Reviewer for the effort in revising the manuscript and for the very helpful suggestions.
1) Ideally all figures shown for all the FXI inhibitors should depict the same parameters for easy comparison. Recommend using FXI activity and aPTT. The axes of the graphs should be the same when describing the same result, example Kaolin-induced aPTT can just be labeled aPTT.
We thank the Reviewer for the suggestions. We have now included only the figures showing the FXIa and aPTT and change the Kaolin-induced aPTT with aPTT
2) When referencing “clotting time” state the method used to analyze, e.g. ROTEM, TEG, aPTT.
We have now indicated the methods utilized for determining the clotting time.
3) After introduction but before discussing the targeting of FXI – consider adding a general description of FXI. Describe structure and function of the factor and include its role in secondary hemostasis. What activates/inhibits, synthesis, etc. Much of this information is interspersed throughout the manuscript and can easily be consolidated into one section for clarity.
We have now added a brief, general description of FXI and its role in the regulation of coagulation cascade (please see pg. 2, at the beginning of paragraph 2).
4) Abelacimab, Mechanisms of action – please condense the detailed description of antibody development. Example – “In 2019 a mAB was developed, using a phage display library, that was capable of binding both FXI and FXIa. The Fab portion exhibited high binding affinity…”
We thank the reviewer for the suggestion. We have now condensate this part as follow: “Abelacimab is a monoclonal antibody capable to bind and inhibit both FXI and FXIa [22]. The Fab portion of abelacimab exhibited high binding affinity to FXI and FXIa and inhibited the 50% of FXIa activity at concentration of 2.8 nM. This effect was shown to be very specific with no inhibitory effect on other human serine protease-type coagulation factors, including factor VIIa, factor IXa, factor Xa, FXIIa, thrombin, and its closest homolog plasma kallikrein [22]. Abelacimab binds with high affinity to the catalytic domain of human FXI and FXIa (Kd of 1.3 ± 0.3 pM and 4.7 ± 2.1 pM, respectively) [22]. The analysis of the X-ray structure of the Fab portion of abelacimab in complex with the FXIa catalytic domain revealed that the antibody traps and stabilizes, an inactive conformation, the protease. By meaning of this activity, abelacimab prolonged the clotting timer, determined by using the aPTT assay, and reduced the amount of thrombin in human plasma in a concentration dependent manner [22]. Figure 1 was also removed.
5) Osocimab section, Figure 3, left panel – the figure copied from reference 24 does not represent changes in PTT, rather it correlates to ROTEM results. Please review and correct the pharmacokinetics section for Osocimab.
In order to combine and summarize data, we decided to exclude figure 3 from the manuscript.
6) The summarized data does not completely match the source data from ref 26. Eg. Summary statistics only available for 8 doses, the drug was essentially undetectable at <0.06 not 0.15 mg/kg. If you include the 2 lower doses (both n=7) that will make figure 4 very messy. Consider selective doses – 0.15, 1.25 and 10 mg/kg – that span the ranges of analyzed doses. If these doses are selected, please update figure 4.
We thank the reviewer and we now indicated that the drug was undetectable ad the dose below 0.06 mg/Kg. We decided to focus our attention to the doses 0,6 1,25 and 2,5 because closely reflected those utilized to the FOXTROT Randomized Clinical Trial.
7) Bay 1831865 section. When describing the dose dependent inhibition of BAY 1831865 there is a description of the dose 3.5 mg but not 75 mg which is shown in figure 5. Is there a reason for including 75 mg in the figure although it is not specifically called out in the description?
We actually partially described the pharmacodynamic profile of 75 mg dose. “Duration of aPTT prolongation was dose-dependent, with ratio-to-baseline values returning to baseline levels between 2 days (7 mg) and 55 days (75 mg and 150 mg) after the infusion (Figure 3).” And “FXI activity returned to baseline between 13 days (3.5 mg) and 55 days (35 mg, 75 mg, and 150 mg).”
8) Please review and the last paragraph on page 10. The first couple of sentences refer to FXI and FXIa activities. I believe the intent was to refer to aPTT and FXIa??
We thank the reviewer for the comment. The sentence on FXI and FXIa is actually correct, as the drug reduced the factor activity.
Also, when referencing “clotting time” (throughout the manuscript, as well as in this paragraph) state the method used to analyze, e.g. ROTEM, TEG, aPTT. The last sentence needs to be clarified with the dose.
We have now indicated that the clotting time was performed using rotational thromboelastometry and clarified the sentence as follow “Clotting time, performed using rotational thromboelastometry, was increased in a dose-dependent manner, with a ratio-to-baseline values ranging from 1.2 (3.5 mg i.v.) to 3.4 after administration of 3.5 mg i.v. and (150 mg i.v., respectively).
9) In the Asundexian section, clearly state that figure 7 is from in vitro spiking studies, not from in vivo studies. The ref on line 344 should be 32. In the pharmacokinetics section, if reporting both FXIa and FXI results then need to describe the difference between FXIa and FXI activity measurements. It’s not clear from the language that one refers to a proprietary testing method and the other to a clinical test method.
We have now deleted figure 7 and add reference 32 (now 35) on line 344. We also indicated the differences between FXIa and FXI activity measurements. “FXIa activity was measured using a fluorogenic substrate, while FXI (clotting) activity was measured using a modified APTT assay”.
10) In the Milvexian section, pages 14-15, there is reference to the high selectivity for FXIa (line 387) but the last sentence in the MOA paragraph states that the molecule exhibited >5000-fold selectivity for 15 of the 17 proteases tested. This is contradictory and there is no mention of other tested proteases in the paragraph. Please clarify. Please expand pharmacodynamic section to include FXIa inhibition description and figure. The last sentence of the paragraph (line 419) needs to be corrected to remove repetition (magnitude) and clarified to define “exposure”.
We thank the Reviewer for the comment. We have now clarified the selectivity towards different proteases as follow: “Milvexian is a potent small molecule that inhibits the active form of FXIa with high affinity and selectivity [33]. This molecule is one of the first oral FXIa inhibitors being developed as new potential antithrombotic drug. Milvexian interacts to FXIa with a Ki equal to 0.11 nM and shows an aPTT EC1.5× of 0.50 μM [33]. The molecule exhibited exquisite selectivity (>5000-fold) for 15 of the 17 proteases tested, such as plasma kallikrein (400-fold) and chymotrypsin (300-fold) [33]. We could not expand the pharmacodynamic section because there are no figures and/or additional data published. We have corrected the last sentence of the paragraph.
11) In the ASO section, pages 15-18, Line 423 – sentence states that “Feromersen reduced by 50% and 90% hepatic FXI mRNA after 6 weeks of treatment (figure 10)” what are the 2 different reduction % referring to? Presumably 2 doses that have not been defined in the paragraph?? This data is not shown in figure 10.
Thank you for the comment. Indeed, the two values were referring to two doses as follow: “Fesomersen reduced by 50% and 90% hepatic FXI mRNA expression after 6 weeks of treatment with 12 and 40 mg/kg dose, respectively”. In figure 10 (now figure 7) it is shown the effect on FXIa activity that reflects the reduction of the mRNA in response to Fesomersen. We have now indicated the figure in the right sentence “The effect on FXI mRNA was associated to a significant reduction of plasma FXI protein activity with a maximal 75% reduction reached by day 21 at the 40 mg/kg dose (Figure 7).”
12) Line 433 – change to read, “…followed by every other day s.c. loading dose regimen and weekly s.c. maintenance dosing.” Please reword/clarify the sentence starting on line 436, “At this dose, following cessation…”
We hope the sentence has now been clarified as follow “The drug was administered with by i.v. infusion, followed by a loading dose and then a s.c. injection every week [35]. The effect on FXI mRNA was associated to a significant reduction of plasma FXI protein activity with a maximal 75% reduction reached by day 21 at the 40 mg/kg dose (Figure 7). At this dose, following cessation of treatment, a complete FXI plasma activity recovery was still evident after 13 weeks of a treatment-free period. “
13) Please confirm that the data shown in figure 10 is adapted from ref 35, figures and doses do not seem consistent with source data.
We have now cancelled this figure.
14) The pharmacodynamics section needs to be edited for clarity. The data in the ASO section is combined from multiple sources without clear delineation which causes inconsistencies. Multiple doses are mentioned from differing studies yet are referenced back to the same figure 10. There is reference to figure 12 (line 486), I believe this should be corrected to figure 11.
We thank the reviewer for the suggestions. Reference 37 and 38 (now 40 and 41) derived from two different source of data but the study is the same. We have now indicated both references for the same study.
15) In the completed phase 2 trials section, look for opportunities to summarize this data dense section.
We thank the Reviewer for this observation. However, we strongly believe that from a clinical point of view, it is very important to know the type of phase 2 studies, patients included, comparators and main outcomes.
16) Conclusion should include a statement that the association between the laboratory results (e.g. aPTT, FXI activity, ROTEM) and clinical outcomes have not been established.
We have now added the following sentence in the conclusions “Different pharmacological approaches are currently in clinical development to inhibit FXIa and pharmacodynamic studies clearly showed an effective inhibition of this coagulation factor. However, the laboratory results (e.g. aPTT, FXI activity, ROTEM) cannot be considered a valid predictor of positive clinical outcomes”.
17) Tables – general comments. Please define all abbreviations for every table and use consistent reporting of PK parameters, eg t1/2 for table 6 is different from the other tables.
We have now defined all the abbreviations.
Table 1 – Random capitalization of words and the title is not clear. Consider something like “Challenges of oral anticoagulation that may not be relevant in the setting of targeted FXI anticoagulation.”
We have now changed the title in: ” Potential clinical indications for anti-FXI.”
Table 2 – Please review for accuracy! Example Abelacimab phase 1 healthy donors (ANT-003) and patients with AF (ANT-004), reference 23. There were 32 healthy subjects enrolled in ANT-003 and 18 subjects in ANT-004. If the intent is to only include those in the AF group then please update the population column to remove healthy volunteers.
Asudexian, Phase 2, missing population (recent MI). Fesomersen, phase 1, change population description to be consistent with rest of the table “Patients with end stage renal disease on chronic hemodialysis”
We have now corrected the table accordingly to the reviewer indication
Table 9 – the parameter row is too busy, edit so that the dose is mentioned once for Day 1 and Day 78.
We have now simplified the table
18) Other edits for clarity/accuracy.
Page 2, Line 69 – change to “…seen in hemophilia A or B are not typically found in patients with FXI deficiency.”
Page 5, Line 109 – change to “…high binding affinity to FXI and FXIa and inhibited 50% of FXIa activity at…” remove “the”.
Page 5, line 117 – remove “By meaning of this activity”, and change “timer” to “time”.
Page 6, line 124 – Remove “These” from beginning of the sentence.
Page 9, line 253 – should read (table 5).
Page 10, line 262 – Change to “…increase of aPTT compared to baseline (figure 5).”
Page 10, paragraph starting on line 272. This paragraph consists of 2 sentences that state the same findings. Select one.
Page 11, AB023 section does not include ref 29, only ref 30 regarding neonatal Fc receptor.
Page 12, pharmacodynamics paragraph – combine the 2nd and 3rd sentences to eliminate repetition.
Edits have been made. Thank you.